# TDMPNet: Prototype Network with Recurrent Top-Down Modulation for Robust Object Classification under Partial Occlusion

Mingqing Xiao[1][*], Adam Kortylewski[2], Ruihai Wu[1][*], Siyuan Qiao[2], Wei Shen[2], and Alan Yuille[2]

[1] Peking University
[2] Johns Hopkins University
{mingqing_xiao, wuruihai}@pku.edu.cn,
{akortyl1,siyuan.qiao,wshen10}@jhu.edu, alan.l.yuille@gmail.com

**Abstract.** Despite deep convolutional neural networks' great success in object classification, recent work has shown that they suffer from a severe generalization performance drop under occlusion conditions that do not appear in the training data. Due to the large variability of occluders in terms of shape and appearance, training data can hardly cover all possible occlusion conditions. However, in practice we expect models to reliably generalize to various novel occlusion conditions, rather than being limited to the training conditions. In this work, we integrate inductive priors including prototypes, partial matching and top-down modulation into deep neural networks to realize robust object classification under novel occlusion conditions, with limited occlusion in training data. We first introduce prototype learning as its regularization encourages compact data clusters for better generalization ability. Then, a visibility map at the intermediate layer based on feature dictionary and activation scale is estimated for partial matching, whose prior sifts irrelevant information out when comparing features with prototypes. Further, inspired by the important role of feedback connection in neuroscience for object recognition under occlusion, a structural prior, i.e. top-down modulation, is introduced into convolution layers, purposefully reducing the contamination by occlusion during feature extraction. Experiment results on partially occluded MNIST, vehicles from the PASCAL3D+ dataset, and vehicles from the cropped COCO dataset demonstrate the improvement under both simulated and real-world novel occlusion conditions, as well as under the transfer of datasets.

## 1 Introduction

In recent years, deep convolutional neural networks (DCNNs) have achieved great success in computer vision tasks, like image classification [22,19,6] and

---

[*] Work done at Johns Hopkins University.

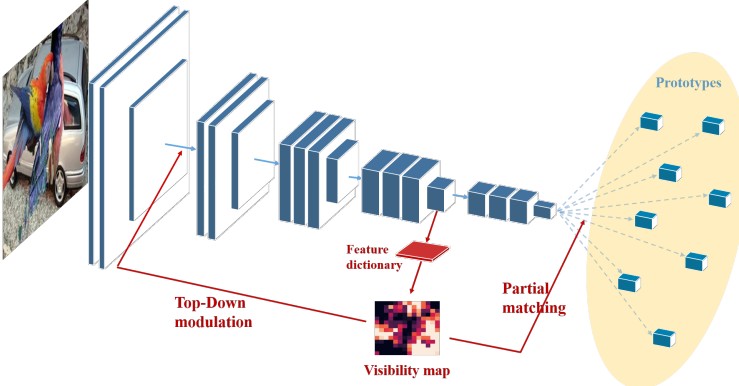

**Fig. 1.** Overall architecture of TDMPNet. We use the convolution layers of VGG-16 as our feature extractor and conduct prototype matching on the features. We estimate a visibility map from the pool-4 layer to focus on target object parts. The visibility map is first used for top-down feedback modulation, reducing the contamination of occlusion during feature extraction, and then for partial matching, sifting irrelevant information out when comparing features and prototypes.

object detection [18,17]. However, widely used deep learning models are not robust under occlusion conditions, especially when ldocclusion does not appear in the training data [3,25,31,8]. While occlusion conditions in accordance with the training data may be solved by e.g. multi-label classification, it is impossible to collect data covering all possible occlusion conditions, and novel occlusion conditions are much tougher to tackle. Over-fitting on the limited training conditions results in failure of generalization to novel occlusion conditions, which can cause fatal consequences in real applications as shown in accidents of driver-assistant systems [2]. In the real world, unexpected occlusion such as a flying tissue in front of objects, which would look like a white box patch on the captured image, always exists. Deep networks can be misguided when they have not seen such a scene in the training data. Humans, on the other hand, are still able to recognize objects under extreme occlusions by unexpected occluders [32]. Therefore, a reliable computer vision model must be robust to novel occlusion other than training conditions. In the following, occlusion refers to **novel occlusion conditions that do not appear in the training data**.

A distribution inconsistency between training and testing data in terms of occlusion causes failures of traditional DCNNs at image classification. Because occlusion patterns are highly variable in terms of appearance and shape, including all possible patterns in the training data is impossible, while biased occlusion patterns do not improve the generalization performance in unbiased conditions [8]. Hence, the inconsistency cannot be avoided and data efficiency regarding occlusion should be considered. Therefore, our work focuses on training a model on limited occlusions while being able to generalize to novel occlusion conditions without assumptions on the occlusion patterns.

There are two main challenges. The first is the over-fitting on the training data, which reduces the generalization ability under novel occlusion conditions. The second is that occluders will contaminate surrounding features during feature extraction. We introduce partial prototype matching to deal with the first problem, and a top-down feedback modulation to tackle the second problem.

In cognitive science, prototype-matching is a popular theory for object recognition. From mathematical perspective, prototypes can be viewed as cluster centers of points from the same class in an embedding space, and distance performs as the matching function. Prototype learning after feature extraction is able to deal with over-fitting [20], as it imposes regularization with a nearest neighbor inductive bias to encourage compact data clusters. Furthermore, different prototypes in one class are able to account for large changes in spatial patterns, such as different viewpoints for 3D objects [8]. Prototypes have been introduced and integrated into deep network structure in few-shot learning task [20] and for rejection and class-incremental learning [27]. However, their distances are simply euclidean distances, which cannot be used directly in occlusion conditions due to the distortion of features in the occluded regions.

To tackle the problem of prototype matching under occlusion conditions, we introduce partial matching with a visibility map to focus on target object parts as illustrated in Figure 1. Wang et al. [24] first discovered that semantic part representations for objects can be found from the internal states of trained DCNNs, based on which Wang et al. [25] and Zhang et al. [31] developed semantic part detection methods. Besides, larger activation scales of internal states are also correlated with objects [30]. Inspired by these works, we employ a filter with a visibility map on possible target object parts based on internal DCNN states and a dictionary to sift out irrelevant information. Experiments show the effective functioning of partial matching according to the filter.

In addition, we propose a top-down feedback modulation with the estimated visibility map (the feedback connection shown in Figure 1) as a structural prior because occluders also contaminate surrounding features during the feature extraction stage. The feedback modulation helps the bottom layers to filter occlusion-induced distortions in the feature activations with high-level information, so that areas around the occluders become less distorted. Our experiments in Section 4 demonstrate the effective contamination reduction. Our top-down modulation is related with some neuroscience conjectures. There are several neuroscience evidence show that recurrent and feedback connections play an important role in object recognition when stimuli are partially occluded [5,15,21,16]. The main conjectures include that the recurrence fills missing data and that it sharpens certain representations by attention refinement [14]. Here we assume that top-down connection could be a neural modulation to filter occlusion-caused anomalous activations.

There are also other techniques that implicitly encourage robustness under occlusion, like cutout regularization [1]. Our model does not conflict with them and can be further combined with these techniques to improve the robustness under novel occlusion conditions.

In summary, this paper makes the following contributions:

– We introduce partial prototype matching with a visibility map based on a feature dictionary into deep neural networks for robust object classification under novel occlusion, with limited occlusion in training data. The prototypes and the visibility map are integrated into a neural network and can be trained end-to-end.
– We further propose a top-down feedback modulation in convolution layers. It imitates the neurological modulation from higher cortex to lower cortex and serves as a structural inductive prior. Experiments show that the feedback effectively reduce the contamination of occlusion during feature extraction.
– Extensive experiments on PASCAL3D+, MNIST, and COCO demonstrate that the proposed model significantly improves the robustness of DCNNs under both simulated and real novel occlusion conditions, as well as under the transfer of datasets. Furthermore, our model can be combined with regularization methods for occlusion-robustness to improve the performance.

## 2    Related Work

**Object classification under partial occlusion**. Fawzi and Frossard [3] have shown that DCNNs are not robust to partial occlusion when inputs are masked out by patches. Devries and Taylor [1] and Yun et al. [29] proposed regularization methods, e.g. cutout, by masking out patches from the images during training, which can improve robustness under occlusion to some extent. Kortylewski et al. [8] proposed dictionary-based Compositional Model. Their model is composed of a traditional DCNN and a compositional model based on the features extracted by DCNN. At runtime, the input is first classified by the DCNN, and will turn to compositional model only when the prediction uncertainty exceeds a threshold, because compositional models are less discriminative than DCNNs. Their model is not end-to-end, does not consider contamination of occlusion during feature extraction and requires a model of occluders. Kortylewski et al. [7] further extended this model to be end-to-end. Differently, our proposed model follows the deep network architecture, reduces influence of occlusions both during and after feature extraction, and is generalizable to novel occlusion conditions.

**Prototype learning in deep networks**. Prototype learning is a classical method in pattern recognition. After the rise of deep neural networks, Yang et al. [27] replace the traditional hand-designed features with features extracted by convolutional neural networks in prototype learning and integrate it into deep networks for both high accuracy and robust pattern classification. Prototypes are also introduced in few-shot and zero-shot learning as part of metric learning [23,20]. Nevertheless, all these works use basic measures like euclidean or cosine distance in prototype matching, which is not suitable for occlusion conditions. We introduce a filter focusing on target object parts to extend prototype matching to occlusion conditions.

**Object part representation inside DCNNs**. Wang et al. [24] found that by clustering feature vectors at different positions from the intermediate layer

of a pre-trained deep neural network, e.g. pool-4 layer in VGG, the patterns of some cluster centers are able to reflect specific object parts. Wang et al. [25] and Zhang et al. [31] use it for semantic part detection, and Kortylewski et al. [8] use it to obtain part components in the compositional model. Related works also include [11], which added a regularizer to encourage the feature representations of DCNNs to cluster during learning, trying to obtain part representations. From another perspective, Zhang et al. [30] tried to encourage each filter to be a part detector by restricting the activations of each filter to be independent, and they estimated the part position by the activation scale. These works demonstrate that object part representation is available inside DCNNs, and activation scale contains information. Based on these ideas, we obtain a filter with a visibility map for partial prototype matching under occlusion by finding possible target object parts with their representations and activation scales, and sifting out other irrelevant information.

**Feedback connections in deep networks**. Despite top-down feedback connection is an ubiquitous structure in biological vision systems, it is not used in typical feed-forward DCNNs. Nayebi et al. [14] has summarized the function conjectures of recurrence in the visual systems and explored possible recurrence structures in CNNs to improve classification performance through architecture search. As for classification task under occlusion, Spoerer et al. [21] explored top-down and lateral connections for digit recognition under occlusion, but their connections are simply convolutional layers without explicit functioning. As for top-down feedback information, Fu et al. [4] learned to focus on smaller areas in the image and Li et al. [10] designed a feedback layer and an emphasis layer. But all of their feedback layers are composed of fully connected layers, which is not interpretable. Some DCNN architectures also borrow the top-down feedback idea, like CliqueNet [28]. Different from these works, our top-down feedback modulation is composed of explainable visibility map focusing on target object parts and is purposefully for reduction in contamination of occlusion.

## 3   Method

Our model is composed of three main parts. The first is prototype learning after feature extraction. Following it is partial matching based on a filter focusing on target object parts to extend prototype matching under occlusion. Finally, top-down modulation is introduced to reduce the contamination of occlusion.

### 3.1   Prototype learning

We conduct prototype learning after feature extraction by DCNNs. Let $x \in \mathbb{R}^{H_0 \times W_0 \times 3}$ denote the input image, our feature extractor is $f_\theta : \mathbb{R}^{H_0 \times W_0 \times 3} \to \mathbb{R}^{H \times W \times C}$, which is composed of convolution layer blocks in typical DCNNs. In contrast to related works [20,27], our feature is a tensor $f_\theta(x) \in \mathbb{R}^{H \times W \times C}$ rather than a vector, in order to maintain spatial information for partial matching in the next section. Suppose there are $N$ classes for classification, we set $M$ prototypes

for each class to account for differences in spatial activation patterns. Therefore prototypes are a set of tensors $p_{i,j} \in \mathbb{R}^{H \times W \times C}$, where $i \in \{1, 2, ..., N\}$ denotes the class of the prototype, and $j \in \{1, 2, ..., M\}$ represents the index in its class.

For feedforward prediction, the image is classified to the class of its nearest prototype according to a distance function $d : \mathbb{R}^{H \times W \times C} \times \mathbb{R}^{H \times W \times C} \rightarrow [0, +\infty)$:

$$Pred(x) = \arg\min_i\{\min_j d(f_\theta(x), p_{i,j})\}. \tag{1}$$

The distance function $d$ can simply be euclidean distance, but experiments in Section 4.2 show that it improves networks slightly due to the contamination of occlusion. A distance for partial matching will be introduced next section.

For backward update of parameters, we use cross entropy loss based on the distances. To be specific, distances between the feature $f_\theta(x)$ and prototypes $p_{i,j}$ produce a probability distribution over classes:

$$Pr(y = k|x) = \frac{\exp(-\gamma d_k)}{\sum_{i=1}^{N} \exp(-\gamma d_i)}, \tag{2}$$

where $d_k = \min_j d(f_\theta(x), p_{k,j})$, and $\gamma$ is a parameter that control the hardness of probability assignment. We set $\gamma$ to be learned by network automatically. Then based on the probability, cross entropy loss is defined:

$$L_{ce}((x, k); \theta, \{p_{i,j}\}) = -\log Pr(y = k|x). \tag{3}$$

Further, a prototype loss is added as the regularization of prototype learning:

$$L_p((x, k); \theta, \{p_{i,j}\}) = \min_{i,j} d(f_\theta(x), p_{i,j}). \tag{4}$$

Different from [27], we only consider the nearest prototype when computing distances and probabilities, because our $M$ prototypes in the same class are designed to represent different states of objects, such as different viewpoints, which may vary a lot in spatial distribution.

We initialize the prototypes by clustering the features of a sub dataset using k-means algorithm [13]. It prevents the degeneration of multiple prototypes to a single prototype.

### 3.2   Partial matching under occlusion

The core problem for extending prototype learning directly to occlusion conditions is the matching function. Since occlusion will contaminate the object feature representation, simple distance between the feature and prototypes won't be valid enough to do classification. Experiments in Section 4.2 show that pure prototype matching improves deep neural networks slightly. Focusing on valid parts in features is required.

We employ a filter with a visibility map based on feature dictionary and activation scale to focus on valid unoccluded parts in features, which enables partial matching. We learn a feature dictionary in the intermediate layer by clustering

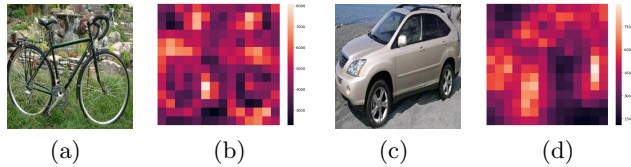

(a)       (b)       (c)       (d)

**Fig. 2.** Visualizaion of activation scale in the pool-4 layer. It shows that the activation scale at parts informative for classification, e.g. object parts, is larger than other areas.

feature vectors over the whole dataset on the feature map, which can represent specific activation patterns of parts in the images. Specifically, feature dictionary is obtained by clustering all normalized vector $v_{k,i,j} \in \mathbb{R}^{1 \times 1 \times C^l}$ at position $(i, j)$ of the feature map $f_\theta^l(x_k) \in \mathbb{R}^{H^l \times W^l \times C^l}$ at the intermediate layer $l$ over the dataset $\{x_k\}$. Related works [24,8] show that cluster centers are mostly activated by similar parts in the images, most of which are object parts. More detailed visualization refer to related works [24,25,8]. Based on the feature dictionary $\{D_k\}$, we compare the similarity between the vectors $f_\theta^l(x)_{i,j}$ of the feature at layer $l$ and each component $D_k$: $S(f_\theta^l(x)_{i,j}, D_k) = \frac{f_\theta^l(x)_{i,j}}{\left\| f_\theta^l(x)_{i,j} \right\|_2} \cdot D_k$. The higher the maximum similarity over $\{D_k\}$ is, the more likely is the area a target object part. Therefore, we can sift occlusion out by its low similarity.

However, there are also a few background activation patterns irrelevant to classification in the dictionary. We use the relative scale of activations to filter them out. As shown in the Figure 2, the scales of activations in a trained network for most irrelevant background are much lower than objects. It is probably because deep networks could learn to focus on image parts that contribute to discrimination most. Considering activation scales is helpful to filter irrelevant background and maintain most informative signals.

Combining the similarity with the feature dictionary and the activation scale enables us to estimate a visibility map that focuses on unoccluded target object parts. The formulation for focusing attention at postion $(i, j)$ in layer $l$ is:

$$a_{i,j}^l = ReLU(\max_k f_\theta(x)_{i,j}^l \cdot D_k). \tag{5}$$

Since the scale of the activation after the ReLU function could be large, we normalize $a_{i,j}^l$. We use the following linear function with clipping since it preserves proper relative relationship among activation scales:

$$A_{i,j}^l = \frac{\min(\max(a_{i,j}^l, a_l), a_u)}{a_u}, \tag{6}$$

where $a_l$ and $a_u$ are lower and upper thresholds that can be dynamically determined according to $\{a_{i,j}^l\}$.

Subsequently, the visibility map is down-sampled to the same spatial scale of $f_\theta(x)_{i,j}$ for partial matching. We let $\{A_{i,j}\}$ denote it. Based on $\{A_{i,j}\}$, partial

matching between the feature and prototypes is enabled. Let $f_\theta(x) \odot A$ denote the application of the filter by scaling vectors $f_\theta(x)_{i,j}$ with $A_{i,j}$. A distance for partial matching under occlusion used for Eq.(1), (2) and (4) is defined as:

$$d(f_\theta(x), p_{i,j}) = \frac{1}{2} \|f_\theta(x) \odot A - p_{i,j} \odot A\|_2^2 \qquad (7)$$

In this way, we only compare unoccluded target object parts based on the estimated visibility map. Due to the high-dimension of $f_\theta(x)$ and $p_{i,j}$, we normalize them on a unit sphere at first and compute the euclidean distance after applying the filter, in order to obtain a valid distance.

We learn the feature dictionary $\{D_k\}$ through clustering. So similar to prototype learning, we initialize it with clustering result on the pre-trained neural network, and add the clustering loss in the whole loss function during training:

$$L_D = \sum_{i,j} \min_k \frac{1}{2} \left\| \frac{f_\theta(x)_{i,j}}{\|f_\theta(x)_{i,j}\|_2} - D_k \right\|_2^2 \qquad (8)$$

Note that we simply add a normalization layer in the network to normalize $d_k$ and ignore the notation in the formula.

### 3.3   Top-Down feedback modulation

Our proposed partial matching only sifts out irrelevant feature vectors when comparing features and prototypes. However, occluders may also contaminate its nearby feature vectors. We propose to filter the occlusion-caused anomalous activitions in the lower layers to reduce such contamination and thus obtain cleaner features around the occluder.

Based on the estimated visibility map at a higher layer, a top-down feedback connection is introduced to reduce the contamination of occlusion in lower layers. Formally, let $\{A_{i,j}^b\}$ denote the up-sampling filter result of $\{A_{i,j}^l\}$ to the same spatial size as the bottom layer b, such as pool-1 layer, and $f_\theta^b$ as the function from input to layer b. A new activation pattern at layer b can be obtained by applying the filter to the old activation:

$$f_\theta^b(x)_{new} = f_\theta^b(x) \odot A^b \qquad (9)$$

The new activation is again feed-forwarded, as a recurrent procedure. The recurrence can be carried out for multiple times, gradually refining features to reduce the contamination of occlusion. The upper threshold in Eq.(6) prevents degeneration of the filter attention to only one point, and the lower threshold in Eq.(6) prevents mistaken filtration due to the possible contamination of occlusion from the bottom layer to top layers.

In summary, the overall architecture with our three components is shown in Figure 1. Our overall loss function for training is:

$$L = L_{ce}((x, k); \theta, \{p_{i,j}\}) + \lambda_1 L_p((x, k); \theta, \{p_{i,j}\}) + \lambda_2 L_D \qquad (10)$$

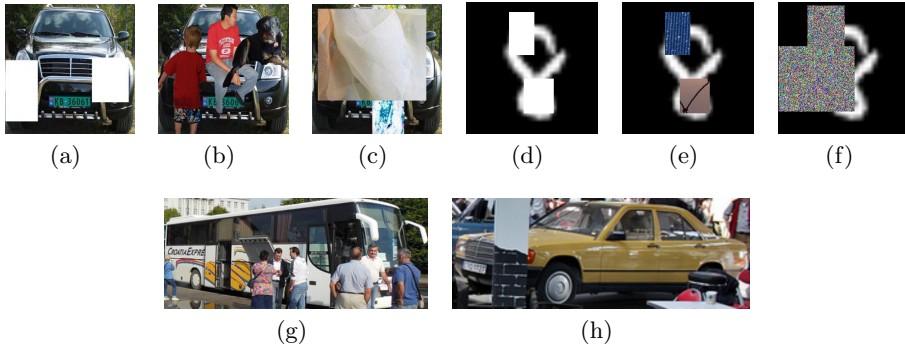

(a) (b) (c) (d) (e) (f)

(g) (h)

**Fig. 3.** Examples of synthetic occlusion and real-world occlusion. (a), (b), (c) correspond to level 1-3 on PASCAL3D+; (d), (e), (f) correspond to level 1-3 on MNIST. Different types of occlusion appearances are: white boxes (a&d), random noise (f), textures (c&e), and natural objects (b). (g) and (h) are real-world occlusion conditions from the COCO dataset.

## 4 Experiments

### 4.1 Dataset and settings

We evaluate our model for object classification on partially occluded MNIST digits [9], vehicles from the PASCAL3D+ dataset [26], and vehicles from the COCO dataset [12]. For PASCAL3D+ and MNIST, we simulate novel occlusion, while for COCO, we split non-occlusion images and occlusion images, and test both direct generalization from PASCAL3D+ to novel occlusion in COCO and performance under training on non-occlusion COCO images.

First, to test the generalization ability to novel occlusion conditions in MNIST and PASCAL3D+, we train our model on original images and test under simulation of partial occlusion by masking out patches in the images and filling them with white boxes, random noise or textures following [8], to imitate unexpected occlusion in front of objects like flying tissues. In addition, we also use the images provided in the VehicleSemanticPart dataset [25] for the PASCAL3D+ vehicles, where occlusion was simulated by superimposing segmented objects over the target object. Different occlusion levels are also defined corresponding to the percentage of occlusion over objects based on the object segmentation masks provided in the PASCAL3D+ and threshold segmentation of the MNIST digits. Examples refer to Figure 3. We use the standard splits for the train and test data. For the PASCAL3D+ dataset, we follow the setup in [24] and [8], that is the task is to discriminate between 12 objects during training, while at test time the 6 vehicle categories are tested.

Then, we crop the COCO dataset with the bounding box ground truth and divide the occluded and unoccluded images manually, whose categories accord with the setup of PASCAL3D+ above. We first directly employ the model trained on PASCAL3D+ on the occluded images from COCO (occluders do not appear

**Table 1.** Classification results for PASCAL3D+ and MNIST with different levels of occlusion (0%, 20-40%, 40-60%, 60-80% of the object are occluded) and different types of occlusion (w = white boxes, n = noise boxes, t = textured boxes, o = natural objects). PrototypeNet denotes only replacing fully-connected layers in VGG by prototype learning, without top-down modulation. Without partial matching denotes simply use euclidean distance for prototype matching. All prototype numbers in one class are set to 4, to be the same as CompDictModel [8]. The best performance is in red font, while the second best is in blue font.

PASCAL3D+ Classification under Occlusion

| Occ. Area | 0% | Level-1: 20-40% | | | | Level-2: 40-60% | | | | Level-3: 60-80% | | | | Mean |
|---|---|---|---|---|---|---|---|---|---|---|---|---|---|---|
| Occ. Type | - | w | n | t | o | w | n | t | o | w | n | t | o | - |
| VGG | 99.4 | 97.5 | 97.5 | 97.3 | 92.1 | 91.7 | 90.6 | 90.2 | 73.0 | 65.0 | 60.7 | 56.4 | 52.2 | 81.8 |
| CompDictModel [8] | 98.3 | 96.8 | 95.9 | 96.2 | 94.4 | 91.2 | 91.8 | 91.3 | 91.4 | 71.6 | 80.7 | 77.3 | 87.2 | 89.5 |
| PrototypeNet without partial matching | 99.2 | 97.1 | 97.6 | 97.2 | 95.3 | 91.2 | 93.0 | 91.3 | 81.3 | 61.9 | 60.9 | 57.9 | 61.5 | 83.5 |
| PrototypeNet with partial matching | 99.3 | 98.4 | 98.9 | 98.5 | 97.3 | 96.4 | 97.1 | 96.2 | 89.2 | 84.0 | 87.4 | 79.7 | 74.5 | 92.1 |
| TDMPNet with 1 recurrence | 99.3 | 98.4 | 98.9 | 98.7 | 97.2 | 96.1 | 97.4 | 96.4 | 90.2 | 81.1 | 87.6 | 81.2 | 76.8 | 92.3 |
| TDMPNet with 2 recurrence | 99.2 | 98.5 | 98.8 | 98.5 | 97.3 | 96.2 | 97.4 | 96.6 | 90.2 | 81.5 | 87.7 | 81.9 | 77.1 | 92.4 |
| TDMPNet with 3 recurrence | 99.3 | 98.4 | 98.9 | 98.5 | 97.4 | 96.1 | 97.5 | 96.6 | 91.6 | 82.1 | 88.1 | 82.7 | 79.8 | 92.8 |
| TDMPNet with 4 recurrence | 99.3 | 98.4 | 98.9 | 98.4 | 97.2 | 96.0 | 97.5 | 96.5 | 91.4 | 81.5 | 87.7 | 82.4 | 79.3 | 92.7 |
| VGG + cutout [1] | 99.4 | 98.1 | 97.9 | 98.2 | 93.8 | 94.8 | 92.3 | 92.4 | 81.3 | 75.4 | 67.7 | 66.3 | 64.8 | 86.3 |
| TDMPNet + cutout | 99.3 | 98.8 | 98.9 | 98.8 | 97.5 | 97.7 | 97.9 | 97.2 | 91.9 | 88.2 | 90.2 | 84.7 | 80.5 | 94.0 |
| Human [8] | 100.0 | 100.0 | | | | 100.0 | | | | 98.3 | | | | 99.5 |

MNIST Classification under Occlusion

| Occ. Area | 0% | Level-1: 20-40% | | | Level-2: 40-60% | | | Level-3: 60-80% | | | Mean |
|---|---|---|---|---|---|---|---|---|---|---|---|
| Occ. Type | - | w | n | t | w | n | t | w | n | t | - |
| VGG | 99.4 | 76.8 | 63.1 | 71.4 | 51.1 | 41.9 | 43.2 | 24.9 | 25.7 | 23.5 | 52.1 |
| CompDictModel [8] | 99.1 | 85.2 | 82.3 | 83.4 | 72.4 | 71.0 | 72.8 | 45.3 | 41.2 | 43.0 | 69.4 |
| PrototypeNet without partial matching | 99.3 | 81.0 | 71.8 | 77.4 | 53.4 | 44.4 | 50.4 | 27.4 | 28.3 | 29.9 | 56.3 |
| PrototypeNet with partial matching | 99.4 | 86.3 | 78.8 | 82.9 | 67.3 | 56.1 | 59.7 | 43.6 | 36.8 | 37.6 | 64.9 |
| TDMPNet with 1 recurrence | 99.4 | 87.6 | 81.4 | 85.3 | 69.3 | 57.9 | 64.0 | 46.1 | 36.8 | 42.1 | 67.0 |
| TDMPNet with 2 recurrence | 99.4 | 88.2 | 82.2 | 85.5 | 70.6 | 59.8 | 64.9 | 47.0 | 38.8 | 42.8 | 67.9 |
| TDMPNet with 3 recurrence | 99.4 | 88.7 | 82.9 | 85.7 | 71.4 | 60.2 | 65.2 | 47.8 | 38.7 | 42.8 | 68.3 |
| TDMPNet with 4 recurrence | 99.5 | 89.3 | 84.2 | 86.3 | 72.7 | 61.6 | 66.3 | 49.3 | 40.0 | 44.0 | 69.3 |
| VGG + cutout [1] | 99.4 | 91.5 | 75.8 | 82.0 | 78.8 | 59.0 | 60.4 | 50.4 | 40.6 | 37.0 | 67.5 |
| TDMPNet + cutout | 99.4 | 92.2 | 95.4 | 93.5 | 79.7 | 84.1 | 78.5 | 57.7 | 59.0 | 51.4 | 79.1 |
| Human [8] | 100.0 | 92.7 | | | 91.3 | | | 64 | | | 84.4 |

in PASCAL3D+), to evaluate the generalization ability under real novel occlusion and transferred datasets. Then, we train our model on non-occlusion images from COCO and test on occluded images. We enrich the training data with PASCAL3D+ due to the insufficiency of images. The occlusion examples are shown in Figure 3.

We utilize convolution layers in a VGG-16 pre-trained on the ImageNet dataset as the feature extraction part. Prototype learning is conducted on the pool-5 layer. The visibility map is estimated from the pool-4 layer, and the top-down modulation is imposed on pool-1 layer. We set feature dictionary components to be 512 for all datasets and use von Mises-Fisher clustering result [8] as the initialization. Other training details refer to the Supplementary Material. We compare our model with VGG-16 finetuned on the datasets, dictionary-based Compositional Model [8], and human baseline. We also compare cutout regularization [1] in VGG and our model, which is similar to adding occlusion in the training data as it masks out patches. The hole number and the length of cutout

is set to be 1 and 48. Other training settings follow the previous settings. The recurrence number of TDMPNet is three if unspecified.

## 4.2   Results on simulated novel occlusion

Results for classification at different occlusion levels are shown in Table 1. They show that DCNNs do not generalize well under synthetic novel occlusion. TDMP-Net significantly outperforms VGG in every occlusion conditions and remains about the same accuracy when there's no occlusion. Further augmented by cutout regularization, our model achieves significantly best results.

**Pure prototype learning improves DCNNs slightly.** As shown in the results, direct prototype learning with simple distance function has little improvement. Though it outperforms VGG in some conditions, the improvements are low compared with follow-up results.

**Partial matching plays a crucial role.** As illustrated by the results, partial matching significantly improves the performance. For the mean accuracy over all conditions, it improves 10.3 percent on PASCAL3D+ and 14.5 percent on MNIST compared with VGG. In the low occlusion level on PASCAL3D+, partial matching achieves the best results even without top-down modulation.

**Top-Down modulation works well for severe occlusions.** Top-down recurrence could effectively improve the performance in relatively hard tasks that even human performance drops. As recurrence times goes up, the features are more pure and therefore performance increases. A more detailed analysis is in the following section. With top-down modulation, the finial mean accuracy outperforms VGG 11 percent on PASCAL3D+ and 17.2 percent on MNIST, reflecting its robustness under partial occlusion.

**Combination with other techniques can further improve the performance greatly.** As shown in Table 1, cutout regularization can significantly boost both VGG and TDMPNet. It shows that TDMPNet does not conflict with other occlusion-robust techniques, and their combination can lead to better results. TDMPNet with cutout regularization achieves the best result for robustness under synthetic novel occlusion, with a boost of 7.7 percent on PASCAL3D+ and 11.6 percent on MNIST compared with VGG with cutout regularization.

**Comparison between TDMPNet and CompDictModel.** Dictionary-based Compositional Model [8] is a model that uses both VGG and a compositional model for classification under partial occlusion. Details refer to Related Work and the original paper. Results show that TDMPNet outperforms Comp-DictModel in most conditions except Level-3 'o' condition in PASCAL3D+. A possible reason is that CompDictModel requires a complex model of occlusion. Differently, our model aims at generalization to novel occlusion conditions and makes no assumptions on occlusion. Another reason is that CompDictModel learn compositional models from the pool-4 layer, which may benefit certain conditions. Detailed analysis refer to the Supplementary Material. In addition, our model is end-to-end, with fewer parameters and is simpler in computation compared with CompDictModel.

**Table 2.** Classification results for cropped COCO. Transfer Accuracy is the direct transfer generalization performance from PASCAL3D+ to cropped COCO. Accuracy is the performance when trained on non-occlusion images of COCO with supplementary images from PASCAL3D+, and tested on occlusion images of COCO.

| Model | Transfer Accuracy | Accuracy |
|---|---|---|
| VGG | 86.66 | 87.27 |
| VGG + cutout | 86.22 | 88.58 |
| TDMPNet | 86.92 | 89.45 |
| TDMPNet + cutout | 87.88 | 90.32 |

### 4.3    Results on real-world novel occlusion

Table 2 are the results of transfer generalization from PASCAL3D+ to novel occlusion in cropped COCO dataset, and the results of training on non-occlusion images from COCO with supplements. It shows actual improvement of TDMPNet under real-world novel occlusion, even under the transfer of datasets. As shown in Table 2, under the transfer generalization, TDMPNet outperforms VGG by 0.26 percent and TDMPNet with cutout demonstrate a more considerable improvement with 1.22 percent accuracy boost, while VGG with cutout do not improve the performance. When trained on unoccluded images from COCO, TDMPNet still demonstrate its superiority over VGG, with a boost of 2.18 percent both without cutout and 1.74 percent both with cutout. Note that VGG with cutout regularization does not generalize its improvement to transferred datasets, while TDMPNet maintains the superiority. It demonstrates the better generalization ability of TDMPNet under novel real-world occlusion conditions and transferred datasets.

### 4.4    Comparison of prototype number

In the previous experiments, we set prototype number as 4 to compare with CompDictModel. We further compare different prototype numbers and visualize images that are assigned to the same prototype.

   **Multiple prototypes improve the performance**. As shown in Table 3, 4 prototypes outperform 1 prototype, while 8 prototypes are about the same as 4 prototypes. It implies that modeling different spatial patterns enables prototypes to be more inclusive, and 4 prototypes are enough to account for the spatial variance in the PASCAL3D+ dataset.

   **Multiple prototypes maintain spatial structures**. As shown in the Supplementary Material, the four prototypes in our model mainly correspond to different viewpoints with certain spatial structure. When there is only one prototype for each class, network simply learns a metric to push all possible activation patterns at a position close with each other, to ensure that the prototype is the closest to all entities. This may lose spatial structure of objects. Multiple prototypes are able to tackle such problem effectively.

**Table 3.** Comparison of different prototype numbers for TDAPNet on PASCAL3D+. The best performance is in red font, while the second best is in blue font.

PASCAL3D+ Classification under Occlusion

| Occ. Area | 0% | Level-1: 20-40% | | | | Level-2: 40-60% | | | | Level-3: 60-80% | | | | Mean |
|---|---|---|---|---|---|---|---|---|---|---|---|---|---|---|
| Occ. Type | - | w | n | t | o | w | n | t | o | w | n | t | o | - |
| 1 prototype, 1 recurrence | 99.2 | 97.9 | 98.5 | 97.9 | 96.4 | 95.1 | 96.5 | 95.2 | 88.6 | 79.1 | 84.8 | 77.9 | 75.1 | 90.9 |
| 1 prototype, 2 recurrence | 99.2 | 97.9 | 98.3 | 97.9 | 96.3 | 95.1 | 96.6 | 95.0 | 89.1 | 78.9 | 85.2 | 78.2 | 75.5 | 91.0 |
| 1 prototype, 3 recurrence | 99.0 | 98.0 | 98.3 | 97.8 | 96.5 | 94.7 | 96.3 | 95.3 | 89.5 | 79.7 | 85.2 | 79.0 | 76.9 | 91.2 |
| 4 prototype, 1 recurrence | 99.3 | 98.4 | 98.9 | 98.7 | 97.2 | 96.1 | 97.4 | 96.4 | 90.2 | 81.1 | 87.6 | 81.2 | 76.8 | 92.3 |
| 4 prototype, 2 recurrence | 99.2 | 98.5 | 98.8 | 98.5 | 97.3 | 96.2 | 97.4 | 96.6 | 90.2 | 81.5 | 87.7 | 81.9 | 77.1 | 92.4 |
| 4 prototype, 3 recurrence | 99.3 | 98.4 | 98.9 | 98.5 | 97.4 | 96.1 | 97.5 | 96.6 | 91.6 | 82.1 | 88.1 | 82.7 | 79.8 | 92.8 |
| 8 prototype, 1 recurrence | 99.3 | 98.7 | 98.9 | 98.7 | 97.5 | 96.4 | 97.5 | 96.7 | 89.6 | 81.1 | 87.6 | 80.9 | 74.7 | 92.1 |
| 8 prototype, 2 recurrence | 99.4 | 98.7 | 99.0 | 98.6 | 97.7 | 96.1 | 97.5 | 96.8 | 90.8 | 82.5 | 88.7 | 82.5 | 78.6 | 92.8 |
| 8 prototype, 3 recurrence | 99.3 | 98.6 | 99.1 | 98.6 | 97.6 | 96.2 | 97.5 | 96.7 | 91.4 | 82.4 | 88.2 | 83.0 | 78.6 | 92.9 |

## 4.5  Analysis of the filter functioning

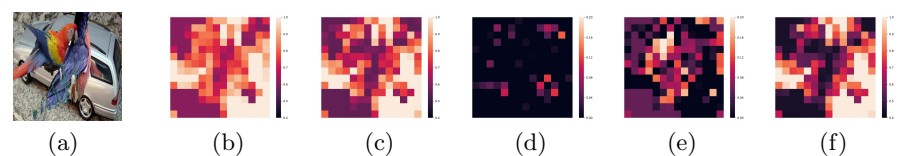

(a)  (b)  (c)  (d)  (e)  (f)

**Fig. 4.** Visualization of visibility maps. Lighter areas represent more focusing attention and darker areas are likely to be filtered. (a) is the occluded image. (b) is the visibility map only based on activation scale. (c) is the visibility map based on feature dictionary and activation scale in the first feed-forward procedure. (d) is the visibility map where feature dictionary enhance attention compared with (b). (e) is the visibility map where feature dictionary reduce attention compared with (b). (f) is the final visibility map after one top-down recurrence.

The classification results demonstrate the importance of the filter with visibility maps for partial matching. We illustrate how the two components in the filter contribute to focusing on informative parts through visualization of visibility maps. As shown in the Figure 4, the activation scale (4(b)) increases the filtering to the background in 4(a). Based on it the feature dictionary further increases the filtering on the occluding parrots (4(e)) and enhance attention on several positions (4(d)), resulting in visibility map 4(c). After a top-down recurrence, the filter further sifts irrelevant information out and mainly focuses on target object parts (4(f)).

## 4.6  Analysis of the top-down modulation effect

We further validate the function of recurrent top-down modulation. It is designed to reduce contamination of occlusion to its surroundings. Therefore, we compare

**Table 4.** Contamination reduction percentage by top-down modulation on PAS-CAL3D+. Larger number reflects better results.

| Occ. Area | Level-1: 20-40% | | | | Level-2: 40-60% | | | | Level-3: 60-80% | | | |
|---|---|---|---|---|---|---|---|---|---|---|---|---|
| Occ. Type | w | n | t | o | w | n | t | o | w | n | t | o |
| TDMPNet with 1 recurrence | 14.1% | 15.2% | 15.7% | 12.5% | 9.5% | 9.2% | 10.1% | 11.7% | 11.1% | 11.1% | 12.9% | 10.9% |
| TDMPNet with 2 recurrence | 16.7% | 17.9% | 18.6% | 13.9% | 10.3% | 10.0% | 10.9% | 13.1% | 13.0% | 13.1% | 15.3% | 12.1% |
| TDMPNet with 3 recurrence | 19.8% | 21.1% | 21.9% | 16.0% | 10.9% | 10.7% | 11.6% | 15.3% | 15.3% | 15.5% | 17.8% | 14.1% |
| TDMPNet with 4 recurrence | 19.9% | 21.4% | 22.2% | 15.8% | 10.6% | 10.4% | 11.4% | 15.3% | 15.6% | 15.8% | 18.2% | 14.2% |

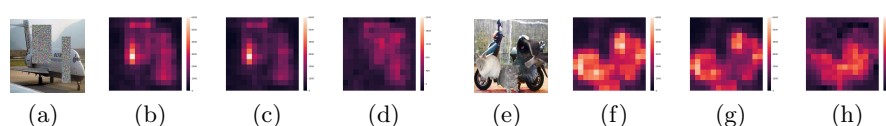

(a)        (b)        (c)        (d)        (e)        (f)        (g)        (h)

**Fig. 5.** Visualization of difference reduction after one top-down recurrence at pool-4 layer. (a)&(e) are the occluded images. (b)&(f) are the activation difference between clean and occluded images before top-down recurrence, while (c)&(g) are the activation difference after top-down recurrence. Lighter areas represent more difference. (d)&(h) are the difference reduction. Lighter areas represent more difference reduction.

the differences between the pool-4 feature of the clean images and the occluded images before and after top-down recurrence. Specifically, let $f_c^0$, $f_c^r$, $f_o^0$, $f_o^r$ denote the pool-4 feature of the clean image before and after recurrence and the occluded image before and after recurrence respectively, and let $m_o$ denote the mask of occlusion area obtained by average down-sampling of the occlusion ground truth. We compute $R_c = 1 - \frac{sum(|f_o^r \odot m_o - f_c^r \odot m_o|)}{sum(|f_o^1 \odot m_o - f_c^1 \odot m_o|)}$ as the contamination reduction percentage. Results in Table 4 clearly show that top-down recurrence is capable of reducing contamination in the bottom layer based on the information from the top layer, and nearly the more the recurrence, the more the reduction. Further, the visualization of difference reduction is in Figure 5, showing the reduction of occlusion-caused difference in features surrounding the occluders.

## 5   Conclusion

In this work, we integrate inductive priors including prototypes, partial matching, and top-down modulation into deep neural networks for robust object classification under novel occlusion conditions, with limited occlusion in training data. The filter in partial matching extends prototype matching to occlusion conditions, and the top-down modulation deals with the contamination of occlusion during feature extraction. Our model significantly improves current deep networks, and its combination with other regularization methods leads to better results. Experiments demonstrate the superiority under both simulated and real novel occlusion conditions and under the transfer of datasets.

**Acknowledgements** This work was partly supported by ONR N00014-18-1-2119.

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
