# OpenReview forum: "TDMPNet: Prototype Network with Recurrent Top-Down Modulation for Robust Object Classification under Partial Occlusion"
_thecvf.com/ECCV/2020/Workshop/VIPriors — VIPriors Poster_

### Official Review · AnonReviewer1 · 2020-07-21
**Occlusion-aware CNN for image classification**

**Confidence:** 3
**Rating:** 7

**Review:**

[Summary] In 2-3 sentences, describe the key ideas, experiments, and their significance.

The authors propose modifying an image classification CNN to add prototype matching with support for partial matches to be more robust to object occlusions. An additional top-down feedback module helps reduce the artefacts caused by occlusions.

[Strengths] What are the strengths of the paper? Clearly explain why these aspects of the paper are valuable.

Extensive experimentation including ablation studies; great amount of analysis to give insight into results.

[Weaknesses] What are the weaknesses of the paper? Clearly explain why these aspects of the paper are weak.

Amount of related works included in comparisons is limited; there are better models than VGG to use as baseline; experiments on real occlusions (i.e. COCO) are limited and complex in setup.

[Overall rating] Paper rating: Accept

[Confidence] /5

[Detailed comments] Additional comments regarding the paper (e.g. typos or other possible improvements you would like to see for the camera-ready version of the paper, if any.)

- Speculation: line 295-296
- Line 437: details on "insufficiency of images" are needed
- Table 2: replace numbered accuracies with some description in the table header
- Grammar: line 119, 285-286
- Typo: line 437 "occulded"
- Line 528: table reference broken
- Multiple places: "Experiments will show" -> "Experiments in section x show", or alternatively consider reorganizing the paper to not (have to) refer to future sections.

---

### Official Review · AnonReviewer2 · 2020-07-28
**TDMPNet: Prototype Network with Recurrent Top-Down Modulation for Robust Object Classification under Partial Occlusion**

**Confidence:** 3
**Rating:** 7

**Review:**

1. [Summary] In 2-3 sentences, describe the key ideas, experiments, and their significance.

 The paper describes an object classification pipeline to better generalize in difficult occluded scenarios. It is composed of three main concepts: prototype learning, partial matching and top-down modulation. Experiments show the benefits of the pipeline at different levels of occlusion in a simulated scenario, as well as in a real scenario.

2. [Strengths] What are the strengths of the paper? Clearly explain why these aspects of the paper are valuable.

 -	The paper is well motivated and clear.
 -	The incremental experimental set-up helps understand the benefits of each of the concepts.
 -	In the concrete experimental set-up authors propose (vehicle images), the pipeline works.
 -	The paper clearly studies how to include visual inductive priors to the network to improve generalization performance.


3. [Weaknesses] What are the weaknesses of the paper? Clearly explain why these aspects of the paper are weak.

 -	The paper seems to have a very controlled experimental set-up, using only vehicles. It would be interesting to study how the method performs with different objects.
 -	Additionally, a comparison with state-of-the-art approaches in this task would add value to the paper.
 -	Lines 440-451:
	- Why 512 dictionary components? Why does it work for all datasets?
	- Have authors tried to use a different model, apart from VGG?


4. [Overall rating] Paper rating.

 7


5. [Justification of rating] Please explain how the strengths and weaknesses aforementioned were weighed in for the rating.

 Readability and use of inductive priors, very positive. Weaknesses are further questions to the paper.


6. [Detailed comments] Additional comments regarding the paper (e.g. typos or other possible improvements you would like to see for the camera-ready version of the paper, if any.)

 -	Line 528: Table ??
 -	Tables are extremely small. Is there any way to change this?
 -	Line 533-539: Shape of prototypes. This would be quite interesting to have it in the main paper, not in the supplementary material

---

### Decision · Program_Chairs · 2020-07-29

**Decision:**

Accept (Poster)

**Comment:**

It is our pleasure to inform you that your paper has been accepted to the poster track of 1st Visual Inductive Priors for Data-Efficient Deep Learning Workshop.

Please note the following deadlines:
* August 11, 2020 - workshop material, including:
 * paper in PDF format;
 * pre-recorded video presentation;
 * slides of the presentation in PDF.
* September 15, 2020 - camera-ready paper

The reviews can be found on OpenReview. Please take these comments and suggestions into account when preparing the camera-ready version of your paper, which is due September 15, 2020. The camera-ready paper should be uploaded to OpenReview.

As part of the workshop, each accepted paper must submit a pre-recorded 90 second talk before August 11, 2020. You will receive more information on how to upload the material shortly. The requirements for the video are:
* Duration: maximum 90 seconds
* MP4 format
* File size max. 100 MB
* Has an inset with a video of the speaker
* 16:9 aspect ratio (strongly preferred)
* 1920x1080 resolution (strongly preferred, at least 720 height)

Our suggested software for pre-recording your presentation is Zoom. For more information, please refer to the following guides:
How to record with Zoom Guide: http://homepages.inf.ed.ac.uk/rbf/ECCV2020HowtoRecordusingZoom.pdf
How to Record with Zoom tutorial: https://www.youtube.com/watch?v=CR199W7HdC0
Please ensure that at least one of the authors of the paper is available to attend the workshop during the allotted times. Note that the workshop will take place in two sessions spread across time zones (details are to follow). We will send instructions on how to connect to the workshop as soon as possible. The schedule for all talks and papers will be posted soon at the workshop website: https://vipriors.github.io.

We look forward to seeing you at the workshop!